# Hyperbolic Graph Neural Networks

**Qi Liu,**[*] **Maximilian Nickel and Douwe Kiela**
Facebook AI Research
{qiliu,maxn,dkiela}@fb.com

## Abstract

Learning from graph-structured data is an important task in machine learning and artificial intelligence, for which Graph Neural Networks (GNNs) have shown great promise. Motivated by recent advances in geometric representation learning, we propose a novel GNN architecture for learning representations on Riemannian manifolds with differentiable exponential and logarithmic maps. We develop a scalable algorithm for modeling the structural properties of graphs, comparing Euclidean and hyperbolic geometry. In our experiments, we show that hyperbolic GNNs can lead to substantial improvements on various benchmark datasets.

## 1 Introduction

We study the problem of supervised learning on entire graphs. Neural methods have been applied with great success to (semi) supervised node and edge classification [26, 51]. They have also shown promise for the classification of graphs based on their structural properties [e.g., 18]. By being invariant to node and edge permutations [3], GNNs can exploit symmetries in graph-structured data, which makes them well-suited for a wide range of problems, ranging from quantum chemistry [18] to modelling social and interaction graphs [50].

In this work, we are concerned with the representational geometry of GNNs. Results in network science have shown that hyperbolic geometry in particular is well-suited for modeling complex networks. Typical properties such as heterogeneous degree distributions and strong clustering can often be explained by assuming an underlying hierarchy which is well captured in hyperbolic space [28, 35]. These insights led, for instance, to hyperbolic geometric graph models, which allow for the generation of random graphs with real-world properties by sampling nodes uniformly in hyperbolic space [1]. Moreover, it has recently been shown that hyperbolic geometry lends itself particularly well for learning hierarchical representations of symbolic data and can lead to substantial gains in representational efficiency and generalization performance [32].

Motivated by these results, we examine if graph neural networks may be equipped with geometrically appropriate inductive biases for capturing structural properties, e.g., information about which nodes are highly connected (and hence more central) or the overall degree distribution in a graph. For this purpose, we extend graph neural networks to operate on Riemannian manifolds with differentiable exponential and logarithmic maps. This allows us to investigate non-Euclidean geometries within a general framework for supervised learning on graphs – independently of the underlying space and its curvature. Here, we compare standard graph convolutional networks [26] that work in Euclidean space with different hyperbolic graph neural networks (HGNNs): one that operates on the Poincaré ball as in [32] and one that operates on the Lorentz model of hyperbolic geometry as in [33]. We focus specifically on the ability of hyperbolic graph neural networks to capture structural properties.

Our contributions are as follows: we generalize graph neural networks to be manifold-agnostic, and show that hyperbolic graph neural networks can provide substantial improvements for full-graph

---

[*]Work done as an AI Resident.

classification. Furthermore, we show that HGNNs are more efficient at capturing structural properties of synthetic data than their Euclidean counterpart; that they can more accurately predict the chemical properties of molecules; and that they can predict extraneous properties of large-scale networks, in this case price fluctuations of a blockchain transaction graph, by making use of the hierarchical structure present in the data. Code and data are available at `https://github.com/facebookresearch/hgnn`

## 2 Related Work

Graph neural networks (GNNs) have received increased attention in machine learning and artificial intelligence due to their attractive properties for learning from graph-structured data [7]. Originally proposed by [19, 41] as a method for learning node representations on graphs using neural networks, this idea was extended to convolutional neural networks using spectral methods [9, 13] and the iterative aggregation of neighbor representations [26, 34, 45]. [22] showed that graph neural networks can be scaled to large-scale graphs. Due to their ability to learn inductive models of graphs, GNNs have found promising applications in molecular fingerprinting [14] and quantum chemistry [18].

There has been an increased interest in hyperbolic embeddings due to their ability to model data with latent hierarchies.

[32] proposed Poincaré for learning hierarchical representations of symbolic data. Furthermore, [33] showed that the Lorentz model of hyperbolic geometry has attractive properties for stochastic optimization and leads to substantially improved embeddings, especially in low dimensions. [16] extended Poincaré embeddings to directed graphs using hyperbolic entailment cones. The representation trade-offs for hyperbolic embeddings were analyzed in [12], which also proposed a combinatorial algorithm to compute embeddings.

Ganea et al. [17] and Gulcehre et al. [21] proposed hyperbolic neural networks and hyperbolic attention networks, respectively, with the aim of extending deep learning methods to hyperbolic space. Our formalism is related to the former in that layer transformations are performed in the tangent space. We propose a model that is applicable to any Riemannian manifold with differentiable log/exp maps, which also allows us to easily extend GNNs to the Lorentz model[2]. Our formalism is related to the latter in that we perform message passing in hyperbolic space, but instead of using the Einstein midpoint, we generalize to any Riemannian manifold via mapping to and from the tangent space.

Hyperbolic geometry has also shown great promise in network science: [28] showed that typical properties of complex networks such as heterogeneous degree distributions and strong clustering can be explained by assuming an underlying hyperbolic geometry and used these insights to develop a geometric graph model for real-world networks [1]. Furthermore, [27, 6, 28] exploited the property of hyperbolic embeddings to embed tree-like graphs with low distortion, for greedy-path routing in large-scale communication networks.

Concurrently with this work, Chami *et al.* [10] also proposed an extension of graph neural networks to hyperbolic geometry. The main difference lies in their attention-based architecture for neighborhood aggregation, which also elegantly supports having trainable curvature parameters at each layer. They show strong performance on link prediction and node classification tasks, and provide an insightful analysis in terms of a graph's $\delta$-hyperbolicity.

## 3 Hyperbolic Graph Neural Networks

Graph neural networks can be interpreted as performing message passing between nodes [18]. We base our framework on graph convolutional networks as proposed in [26], where node representations are computed by aggregating messages from direct neighbors over multiple steps. That is, the message from node $v$ to its receiving neighbor $u$ is computed as $\mathbf{m}_v^{k+1} = \mathbf{W}^k \tilde{\mathbf{A}}_{uv} \mathbf{h}_v^k$. Here $\mathbf{h}_v^k$ is the representation of node $v$ at step $k$, $\mathbf{W}^k \in \mathbb{R}^{h \times h}$ constitutes the trainable parameters for step $k$ (i.e., the $k$-th layer), and $\tilde{\mathbf{A}} = \mathbf{D}^{-\frac{1}{2}}(\mathbf{A} + \mathbf{I})\mathbf{D}^{-\frac{1}{2}}$ captures the connectivity of the graph. To get $\tilde{\mathbf{A}}$, the identity matrix $\mathbf{I}$ is added to the adjacency matrix $\mathbf{A}$ to obtain self-loops for each node, and the

resultant matrix is normalized using the diagonal degree matrix ($\mathbf{D}_{ii} = \sum_j (\mathbf{A}_{ij} + \mathbf{I}_{ij})$). We then obtain a new representation of $u$ at step $k + 1$ by summing up all the messages from its neighbors before applying the activation function $\sigma$: $\mathbf{h}_u^{k+1} = \sigma(\sum_{v \in \mathcal{I}(u)} \mathbf{m}_v^{t+1})$, where $\mathcal{I}(u)$ is the set of in-neighbors of $u$, i.e. $v \in \mathcal{I}(u)$ if and if only $v$ has an edge pointing to $u$. Thus, in a more compact notation, the information propagates on the graph as:

$$\mathbf{h}_u^{k+1} = \sigma \left( \sum_{v \in \mathcal{I}(u)} \tilde{\mathbf{A}}_{uv} \mathbf{W}^k \mathbf{h}_v^k \right). \tag{1}$$

### 3.1 Graph Neural Networks on Riemannian Manifolds

A graph neural network comprises a series of basic operations, i.e. message passing via linear maps and pointwise non-linearities, on a set of nodes that live in a given space. While such operations are well-understood in Euclidean space, their counterparts in non-Euclidean space (which we are interested in here) are non-trivial. We generalize the notion of a graph convolutional network such that the network operates on Riemannian manifolds and becomes agnostic to the underlying space. Since the tangent space of a point on Riemannian manifolds always is Euclidean (or a subset of Euclidean space), functions with trainable parameters are executed there. The propagation rule for each node $u \in V$ is calculated as:

$$\mathbf{h}_u^{k+1} = \sigma \left( \exp_{\mathbf{x}'}( \sum_{v \in \mathcal{I}(u)} \tilde{\mathbf{A}}_{uv} \mathbf{W}^k \log_{\mathbf{x}'}(\mathbf{h}_v^k)) \right). \tag{2}$$

At layer $k$, we map each node representation $\mathbf{h}_v^k \in \mathcal{M}$, where $v \in \mathcal{I}(u)$ is a neighbor of $u$, to the tangent space of a chosen point $\mathbf{x}' \in \mathcal{M}$ using the logarithmic map $\log_{\mathbf{x}'}$. Here $\tilde{\mathbf{A}}$ and $\mathbf{W}^k$ are the normalized adjacency matrix and the trainable parameters, respectively, as in Equation 1. An exponential map $\exp_{\mathbf{x}'}$ is applied afterwards to map the linearly transformed tangent vector back to the manifold.

The activation $\sigma$ is applied after the exponential map to prevent model collapse: if the activation was applied before the exponential map, i.e. $\mathbf{h}_u^{k+1} = \exp_{\mathbf{x}'}\left( \sigma(\sum_{v \in \mathcal{I}(u)} \tilde{\mathbf{A}}_{uv} \mathbf{W}^k \log_{\mathbf{x}'}(\mathbf{h}_v^k)) \right)$, the exponential map $\exp_{\mathbf{x}'}$ at step $k$ would have been cancelled by the logarithmic map $\log_{\mathbf{x}'}$ at step $k + 1$ as $\log_{\mathbf{x}'}(\exp_{\mathbf{x}'}(\mathbf{h})) = \mathbf{h}$. Hence, any such model would collapse to a vanilla Euclidean GCN with a logarithmic map taking the input features of the GCN and an exponential map taking its outputs. An alternative to prevent such collapse would be to introduce bias terms as in [17]. Importantly, when applying the non-linearity directly on a manifold $\mathcal{M}$, we need to ensure that its application is manifold preserving, i.e., that $\sigma : \mathcal{M} \to \mathcal{M}$. We will propose possible choices for non-linearities in the discussion of the respective manifolds.

### 3.2 Riemannian Manifolds

A Riemannian manifold $(\mathcal{M}, g)$ is a real and smooth manifold equipped with an inner product $g_{\mathbf{x}} : \mathcal{T}_{\mathbf{x}}\mathcal{M} \times \mathcal{T}_{\mathbf{x}}\mathcal{M} \to \mathbb{R}$ at each point $\mathbf{x} \in \mathcal{M}$, which is called a Riemannian metric and allows us to define the geometric properties of a space such as angles and the length of a curve.

We experiment with Euclidean space and compare it to two different hyperbolic manifolds (note that there exist multiple equivalent hyperbolic models, such as the Poincaré ball and the Lorentz model, for which transformations exist that preserve all geometric properties including isometry).

**Euclidean Space**  The Euclidean manifold is a manifold with zero curvature. The metric tensor is defined as $g^E = \operatorname{diag}([1, 1, \ldots, 1])$. The closed-form distance, i.e. the length of the geodesic, which is a straight line in Euclidean space, between two points is given as:

$$d(\mathbf{x}, \mathbf{y}) = \sqrt{\sum_i (x_i - y_i)^2} \tag{3}$$

The exponential map of the Euclidean manifold is defined as:

$$\exp_{\mathbf{x}}(\mathbf{v}) = \mathbf{x} + \mathbf{v} \tag{4}$$

The logarithmic map is given as:

$$\log_{\mathbf{x}}(\mathbf{y}) = \mathbf{y} - \mathbf{x} \tag{5}$$

In order to make sure that the Euclidean manifold formulation is equivalent to the vanilla GCN model described in Equation 1, as well as for reasons of computational efficiency, we choose $\mathbf{x}' = \mathbf{x}_0$ (i.e., the origin) as the fixed point on the manifold in whose tangent space we operate.

**Poincaré Ball Model** The Poincaré ball model with constant negative curvature corresponds to the Riemannian manifold $(\mathcal{B}, g_{\mathbf{x}}^{\mathcal{B}})$, where $\mathcal{B} = \{\mathbf{x} \in \mathbb{R}^n : \|\mathbf{x}\| < 1\}$ is an open unit ball. Its metric tensor is $g_{\mathbf{x}}^{\mathcal{B}} = \lambda_{\mathbf{x}}^2 g^E$, where $\lambda_{\mathbf{x}} = \frac{2}{1-\|\mathbf{x}\|^2}$ is the conformal factor and $g^E$ is the Euclidean metric tensor (see above). The distance between two points $\mathbf{x}, \mathbf{y} \in \mathcal{B}$ is given as:

$$d_{\mathcal{B}}(\mathbf{x}, \mathbf{y}) = \operatorname{arcosh}\left(1 + 2\frac{\|\mathbf{x} - \mathbf{y}\|^2}{(1 - \|\mathbf{x}\|^2)(1 - \|\mathbf{y}\|^2)}\right). \tag{6}$$

For any point $\mathbf{x} \in \mathcal{B}$, the exponential map $\exp_{\mathbf{x}} : \mathcal{T}_{\mathbf{x}}\mathcal{B} \to \mathcal{B}$ and the logarithmic map $\log_{\mathbf{x}} : \mathcal{B} \to \mathcal{T}_{\mathbf{x}}\mathcal{B}$ are defined for the tangent vector $\mathbf{v} \neq \mathbf{0}$ and the point $\mathbf{y} \neq \mathbf{0}$, respectively, as:

$$\exp_{\mathbf{x}}(\mathbf{v}) = \mathbf{x} \oplus \left(\tanh\left(\frac{\lambda_{\mathbf{x}}\|\mathbf{v}\|}{2}\right)\frac{\mathbf{v}}{\|\mathbf{v}\|}\right)$$

$$\log_{\mathbf{x}}(\mathbf{y}) = \frac{2}{\lambda_{\mathbf{x}}}\operatorname{arctanh}(\| - \mathbf{x} \oplus \mathbf{y}\|)\frac{-\mathbf{x} \oplus \mathbf{y}}{\| - \mathbf{x} \oplus \mathbf{y}\|}, \tag{7}$$

where $\oplus$ is the Möbius addition for any $\mathbf{x}, \mathbf{y} \in \mathcal{B}$:

$$\mathbf{x} \oplus \mathbf{y} = \frac{(1 + 2\langle\mathbf{x}, \mathbf{y}\rangle + \|\mathbf{y}\|^2)\mathbf{x} + (1 - \|\mathbf{x}\|^2)\mathbf{y}}{1 + 2\langle\mathbf{x}, \mathbf{y}\rangle + \|\mathbf{x}\|^2\|\mathbf{y}\|^2} \tag{8}$$

Similar to the Euclidean case, and following [17], we use $\mathbf{x}' = \mathbf{x}_0$. On the Poincaré ball, we employ pointwise non-linearities which are norm decreasing, i.e., where $|\sigma(x)| \leq |x|$ (which is true for e.g. ReLU and leaky ReLU). This ensures that $\sigma : \mathbb{B} \to \mathbb{B}$ since $\|\sigma(\mathbf{x})\| \leq \|\mathbf{x}\|$.

**Lorentz Model** The Lorentz model avoids numerical instabilities that may arise with the Poincaré distance (mostly due to the division) [33]. Its stability is particularly useful for our architecture, since we have to apply multiple sequential exponential and logarithmic maps in deep GNNs, which would normally compound numerical issues, but which the Lorentz model avoids. Let $\mathbf{x}, \mathbf{y} \in \mathbb{R}^{n+1}$, then the Lorentzian scalar product is defined as:

$$\langle\mathbf{x}, \mathbf{y}\rangle_{\mathcal{L}} = -x_0 y_0 + \sum_{i=1}^{n} x_n y_n \tag{9}$$

The Lorentz model of n-dimensional hyperbolic space is then defined as the Riemannian manifold $(\mathcal{L}, g_{\mathbf{x}}^{\mathcal{L}})$, where $\mathcal{L} = \{\mathbf{x} \in \mathbb{R}^{n+1} : \langle\mathbf{x}, \mathbf{x}\rangle_{\mathcal{L}} = -1, x_0 > 0\}$ and where $g^{\mathcal{L}} = \operatorname{diag}([-1, 1, \ldots, 1])$. The induced distance function is given as:

$$d_{\mathcal{L}}(\mathbf{x}, \mathbf{y}) = \operatorname{arcosh}(-\langle\mathbf{x}, \mathbf{y}\rangle_{\mathcal{L}}) \tag{10}$$

The exponential map $\exp_{\mathbf{x}} : \mathcal{T}_{\mathbf{x}}\mathcal{L} \to \mathcal{L}$ and the logarithmic map $\log_{\mathbf{x}} : \mathcal{L} \to \mathcal{T}_{\mathbf{x}}\mathcal{L}$ are defined as:

$$\exp_{\mathbf{x}}(\mathbf{v}) = \cosh(\|\mathbf{v}\|_{\mathcal{L}})\mathbf{x} + \sinh(\|\mathbf{v}\|_{\mathcal{L}})\frac{\mathbf{v}}{\|\mathbf{v}\|_{\mathcal{L}}}$$

$$\log_{\mathbf{x}}(\mathbf{y}) = \frac{\operatorname{arcosh}(-\langle\mathbf{x}, \mathbf{y}\rangle_{\mathcal{L}})}{\sqrt{\langle\mathbf{x}, \mathbf{y}\rangle_{\mathcal{L}}^2 - 1}}(\mathbf{y} + \langle\mathbf{x}, \mathbf{y}\rangle_{\mathcal{L}}\mathbf{x}), \tag{11}$$

where $\|\mathbf{v}\|_{\mathcal{L}} = \sqrt{\langle\mathbf{v}, \mathbf{v}\rangle_{\mathcal{L}}}$.

The origin, i.e., the zero vector in Euclidean space and the Poincaré ball, is equivalent to $(1, 0, \ldots, 0)$ in the Lorentz model, which we use as $\mathbf{x}'$. Since activation functions such as ReLU and leaky ReLU are not manifold-preserving in the Lorentz model, we first use Equation 12 to map the point from Lorentz to Poincaré and apply the activation $\sigma$, before mapping it back using Equation 13:

$$p_{\mathcal{L}\to\mathcal{B}}(x_0, x_1, \ldots, x_n) = \frac{(x_1, \ldots, x_n)}{x_0 + 1} \tag{12}$$

$$p_{\mathcal{B}\to\mathcal{L}}(x_0, x_1, \ldots, x_n) = \frac{(1 + \|\mathbf{x}\|^2, 2x_1, \ldots, 2x_n)}{1 - \|\mathbf{x}\|^2} \tag{13}$$

| | Dimensionality | | | | |
|---|---|---|---|---|---|
| | 3 | 5 | 10 | 20 | 256 |
| Euclidean | $77.2 \pm 0.12$ | $90.0 \pm 0.21$ | $90.6 \pm 0.17$ | $94.8 \pm 0.25$ | $95.3 \pm 0.17$ |
| Poincare | $93.0 \pm 0.05$ | $95.6 \pm 0.14$ | $95.9 \pm 0.14$ | $96.2 \pm 0.06$ | $93.7 \pm 0.05$ |
| Lorentz | $94.1 \pm 0.03$ | $95.1 \pm 0.25$ | $96.4 \pm 0.23$ | $96.6 \pm 0.22$ | $95.3 \pm 0.28$ |

Table 1: F1 (macro) score and standard deviation of classifying synthetically generated graphs according to the underlying graph generation algorithm (high is good).

### 3.3 Centroid-Based Regression and Classification

The output of a hyperbolic graph neural network with $K$ steps consists of a set of node representations $\{\mathbf{h}_1^K, ..., \mathbf{h}_{|V|}^K\}$, where each $\mathbf{h}_i^K \in \mathcal{M}$. Standard parametric classification and regression methods in Euclidean space are not generally applicable in the hyperbolic case. Hence, we propose an extension of the underlying idea of radial basis function networks [8, 36] to Riemannian manifolds. The key idea is to use a differentiable function $\psi : \mathcal{M} \to \mathbb{R}^d$ that can be used to summarize the structure of the node embeddings. More specifically, we first introduce a list of centroids $\mathcal{C} = [\mathbf{c}_1, \mathbf{c}_2, ..., \mathbf{c}_{|\mathcal{C}|}]$, where each $\mathbf{c}_i \in \mathcal{M}$. The centroids are learned jointly with the GNN using backpropagation. The pairwise distance between $\mathbf{c}_i$ and $\mathbf{h}_j^K$ is calculated as: $\psi_{ij} = d(\mathbf{c}_i, \mathbf{h}_j^K)$. Next, we concatenate all distances $(\psi_{1j}, ..., \psi_{|\mathcal{C}|j}) \in \mathbb{R}^{|\mathcal{C}|}$ to summarize the position of $\mathbf{h}_j^K$ relative to the centroids. For node-level regression,

$$\hat{y} = \mathbf{w}_o^T (\psi_{1j}, ..., \psi_{|\mathcal{C}|j}), \qquad (14)$$

where $\mathbf{w}_o \in \mathbb{R}^{|C|}$, and for node-level classification,

$$p(y_j) = \text{softmax} \left( \mathbf{W}_o (\psi_{1j}, ..., \psi_{|\mathcal{C}|j}) \right), \qquad (15)$$

where $\mathbf{W}_o \in \mathbb{R}^{c \times |C|}$ and $c$ denotes the number of classes.

For graph-level predictions, we first use average pooling to combine the distances of different nodes, obtaining $(\psi_1, ..., \psi_{|\mathcal{C}|})$, where $\psi_i = \sum_{j=1}^{|V|} \psi_{ij}/|V|$, before feeding $(\psi_1, ..., \psi_{|\mathcal{C}|})$ into fully connected networks. Standard cross entropy and mean square error are used as loss functions for classification and regression, respectively.

### 3.4 Other details

The input features of neural networks are typically embeddings or features that live in Euclidean space. For Euclidean features $\mathbf{x}^{\mathbf{E}}$, we first apply $\exp_{\mathbf{x}'}(\mathbf{x}^{\mathbf{E}})$ to map it into the Riemannian manifolds. To initialize embeddings $\mathbf{E}$ within the Riemannian manifold, we first uniformly sample from a range (e.g. $[-0.01, 0.01]$) to obtain Euclidean embeddings, before normalizing the embeddings to ensure that each embedding $\mathbf{e}_i \in \mathcal{M}$. The Euclidean manifold is normalized into a unit ball to make sure we compare fairly with the Poincaré ball and the Lorentz model. This normalization causes minor differences with respect to the vanilla GCN model of Kipf & Welling [26] but as we show in the appendix, in practice this does not cause any significant dissimilarities. We use leaky ReLU as the activation function $\sigma$ with the negative slope 0.5. We use RAMSGrad [4] and AMSGrad for hyperbolic parameters and Euclidean parameters, respectively.

## 4 Experiments

In the following experiments, we will compare the performance of models using different spaces within the Riemannian manifold, comparing the canonical Euclidean version to Hyperbolic Graph Neural Networks using either Poincaré or Lorentz manifolds.

### 4.1 Synthetic Structures

First, we attempt to corroborate the hypothesis that hyperbolic graph neural networks are better at capturing structural information of graphs than their Euclidean counterpart. To that end, we design a

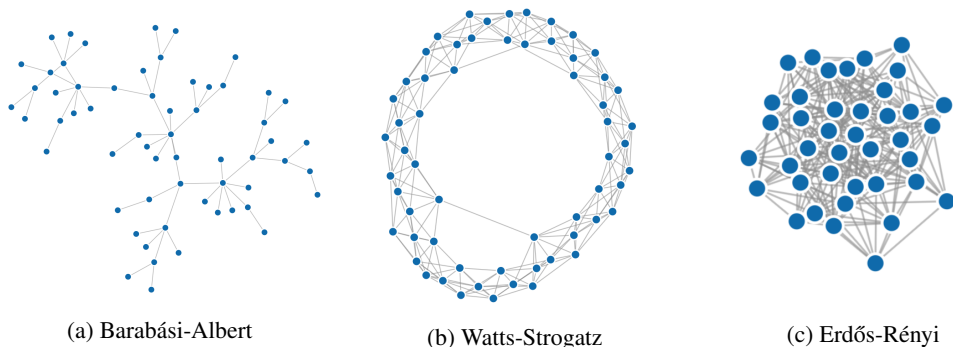

(a) Barabási-Albert             (b) Watts-Strogatz             (c) Erdős-Rényi

Figure 1

synthetic experiment, such that we have full control over the amount of structural information that is required for the classification decision. Specifically, our task is to classify synthetically generated graphs according to the underlying generation algorithm. We choose 3 distinct graph generation algorithms: Erdős-Rényi [15], Barabási-Albert [2] and Watts-Strogatz [46] (see Figure 1).

The graphs are constructed as follows. For each graph generation algorithm we uniformly sample a number of nodes between 100 and 500 and subsequently employ the graph generation algorithm on the nodes. For Barabási-Albert graphs, we set the number of edges to attach from a new node to existing nodes to a random number between 1 and 100. For Erdős-Rényi, the probability for edge creation is set to $0.1 - 1$. For Watts-Strogatz, each node is connected to $1 - 100$ nearest neighbors in the ring topology, and the probability of rewiring each edge is set to $0.1 - 1$.

Table 1 shows the results of classifying the graph generation algorithm (as measured by F1 score over the three classes). For our comparison, we follow [32] and show results for different numbers of dimensions. We observe that our hyperbolic methods outperform the Euclidean alternative by a large margin. Owing to the representational efficiency of hyperbolic methods, the difference is particularly big for low-dimensional cases. The Lorentz model does better than the Poincaré one in all but one case. The differences become smaller with higher dimensionality, as we should expect, but hyperbolic methods still do better in the relatively high dimensionality of 256. We speculate that this is due to their having better inductive biases for capturing the structural properties of the graphs, which is extremely important for solving this particular task.

## 4.2 Molecular Structures

Graphs are ubiquitous as data structures, but one domain where neural networks for graph data have been particularly impactful is in modeling chemical problems. Applications include molecular design [31], fingerprinting [14] and poly pharmaceutical side-effect modeling [52].

Molecular property prediction has received attention as a reasonable benchmark for supervised learning on molecules [18]. A popular choice for this purpose is the QM9 dataset [37]. Unfortunately, it is hard to compare to previous work on this dataset, as the original splits from [18] are no longer available (per personal correspondence). One characteristic with QM9 is that the molecules are relatively small (around 10 nodes per graph) and that there is high variance in the results. Hence, we instead use the much larger ZINC dataset [44, 24, 23], which has been used widely in graph generation for molecules using machine learning methods [25, 31]. However, see the appendix for results on QM8 [40, 39] and QM9 [40, 38].

ZINC is a large dataset of commercially available drug-like chemical compounds. For ZINC, the input consists of embedding representations of atoms together with an adjacency matrix, without any additional handcrafted features. A master node [18] is added to the adjacency matrix to speed up message passing. The dataset consists of 250k examples in total, out of which we randomly sample 25k for the validation and test sets, respectively. On average, these molecules are bigger (23 heavy atoms on average) and structurally more complex than the molecules in QM9. ZINC is multi-relational, i.e. there are four types of relations for molecules, i.e. single bond, double bond, triple bond and aromatic bond.

| | logP | | | | |
|---|---|---|---|---|---|
| | 3 | 5 | 10 | 20 | 256 |
| Euclidean | $6.7 \pm 0.07$ | $4.7 \pm 0.03$ | $4.7 \pm 0.02$ | $3.6 \pm 0.00$ | $3.3 \pm 0.00$ |
| Poincare | $5.7 \pm 0.00$ | $4.6 \pm 0.03$ | $3.6 \pm 0.02$ | $3.2 \pm 0.01$ | $3.1 \pm 0.01$ |
| Lorentz | $5.5 \pm 0.02$ | $4.5 \pm 0.03$ | $3.3 \pm 0.03$ | $2.9 \pm 0.01$ | $2.4 \pm 0.02$ |
| | QED | | | | |
| | 3 | 5 | 10 | 20 | 256 |
| Euclidean | $22.4 \pm 0.21$ | $15.9 \pm 0.14$ | $14.5 \pm 0.09$ | $10.2 \pm 0.08$ | $6.4 \pm 0.06$ |
| Poincare | $22.1 \pm 0.01$ | $14.9 \pm 0.13$ | $10.2 \pm 0.02$ | $6.9 \pm 0.02$ | $6.0 \pm 0.04$ |
| Lorentz | $21.9 \pm 0.12$ | $14.3 \pm 0.12$ | $8.7 \pm 0.04$ | $6.7 \pm 0.06$ | $4.7 \pm 0.00$ |
| | SAS | | | | |
| | 3 | 5 | 10 | 20 | 256 |
| Euclidean | $20.5 \pm 0.04$ | $16.8 \pm 0.07$ | $14.5 \pm 0.11$ | $9.6 \pm 0.05$ | $9.2 \pm 0.08$ |
| Poincare | $18.8 \pm 0.03$ | $16.1 \pm 0.08$ | $12.9 \pm 0.04$ | $9.3 \pm 0.07$ | $8.6 \pm 0.02$ |
| Lorentz | $18.0 \pm 0.15$ | $16.0 \pm 0.15$ | $12.5 \pm 0.07$ | $9.1 \pm 0.08$ | $7.7 \pm 0.06$ |

Table 2: Mean absolute error of predicting molecular properties: the water-octanal partition coefficient (logP); qualitative estimate of drug-likeness (QED); and synthetic accessibility score (SAS). Scaled by 100 for table formatting (low is good).

| | logP | QED | SAS |
|---|---|---|---|
| DTNN [43] | $4.0 \pm 0.03$ | $8.1 \pm 0.04$ | $9.9 \pm 0.06$ |
| MPNN [18] | $4.1 \pm 0.02$ | $8.4 \pm 0.05$ | $9.2 \pm 0.07$ |
| GGNN [29] | $3.2 \pm 0.20$ | $6.4 \pm 0.20$ | $9.1 \pm 0.10$ |
| Euclidean | $3.3 \pm 0.00$ | $6.4 \pm 0.06$ | $9.2 \pm 0.08$ |
| Poincare | $3.1 \pm 0.01$ | $6.0 \pm 0.04$ | $8.6 \pm 0.02$ |
| Lorentz | $2.4 \pm 0.02$ | $4.7 \pm 0.00$ | $7.7 \pm 0.06$ |

Table 3: Mean absolute error of predicting molecular properties logP, QED and SAS, as compared to current state-of-the-art deep learning methods. Scaled by 100 for table formatting (low is good).

First, in order to enable our graph neural networks to handle multi-relational data, we follow [42] and extend Equation 2 to incorporate a multirelational weight matrix $\mathbf{W}_r^k$ where $r \in \mathcal{R}$ is the set of relations, which we sum over. As before, we compare the various methods for different dimensionalities. The results can be found in Table 2.

We also compare to three strong baselines on exactly the same data splits of the ZINC dataset: graph-gated neural networks [GGNN; 29], deep tensor neural networks [DTNN; 43] and message-passing neural networks [MPNN; 18]. GGNN adds a GRU-like update [11] that incorporates information from neighbors and previous timesteps in order to update node representations. DTNN takes as the input a fully connected weighted graph and aggregates node representations using a deep tensor neural network. For DTNN and MPNN we use the implementations in DeepChem[3], a well-known open-source toolchain for deep-learning in drug discovery, materials science, quantum chemistry, and biology. For GGNN, we use the publicly available open-source implementation[4]. A comparison of our proposed approach to these methods can be found in Table 3.

We find that the Lorentz model outperforms the Poincaré ball on all properties, which illustrates the benefits of its improved numerical stability. The Euclidean manifold performs worse than the hyperbolic versions, confirming the effectiveness of hyperbolic models for modeling complex structures in graph data. Furthermore, as can be seen in the appendix, the computational overhead of using non-Euclidean manifolds is relatively minor.

|          | Dev             | Test            |
|----------|-----------------|-----------------|
| Node2vec | 54.10 ± 1.63    | 52.44 ± 1.10    |
| ARIMA    | 54.50 ± 0.16    | 53.07 ± 0.06    |
| Euclidean| 56.15 ± 0.30    | 53.95 ± 0.20    |
| Poincare | 57.03 ± 0.28    | 54.41 ± 0.24    |
| Lorentz  | 57.52 ± 0.35    | 55.51 ± 0.37    |

|      | "Whale" nodes | All nodes |
|------|---------------|-----------|
| Norm | 0.20129       | 0.33178   |

Table 4: Accuracy of predicting price fluctations (up-down) for the Ether/USDT market rate based on graph dynamics.

Table 5: Average norm of influential "whale" nodes. Whales are significantly closer to the origin than average, indicating their importance.

GGNN obtains comparable results to the Euclidean GNN. DTNN performs worse than the other models, as it relies on distance matrices ignoring multi-relational information during message passing.

### 4.3 Blockchain Transaction Graphs

In terms of publicly accessible graph-structured data, blockchain networks like Ethereum constitute some of the largest sources of graph data in the world. Interestingly, financial transaction networks such as the blockchain have a strongly hierarchical nature: the blockchain ecosystem has even invented its own terminology for this, e.g., the market has long been speculated to be manipulated by "whales". A whale is a user (address) with enough financial resources to move the market in their favored direction. The structure of the blockchain graph and its dynamics over time have been used as a way of quantifying the "true value" of a network [49, 47]. Blockchain networks have uncharacteristic dynamics [30], but the distribution of wealth on the blockchain follows a power-law distribution that is arguably (even) more skewed than in traditional markets [5]. This means that the behavior of important "whale" nodes in the graph might be more predictive of fluctations (up or down) in the market price of the underlying asset, which should be easier to capture using hyperbolic graph neural networks.

Here, we study the problem of predicting price fluctations for the underlying asset of the Ethereum blockchain [48], based on the large-scale behavior of nodes in transaction graphs (see the appendix for more details). Each node (i.e., address in the transaction graph) is associated with the same embedding over all timepoints. Models are provided with node embeddings and the transaction graph for a given time frame, together with the Ether/USDT market rate for the given time period. The transaction graph is a directed multi-graph where edge weights correspond to the transaction amount. To encourage message passing on the graphs, we enhance the transaction graphs with inverse edges $u \to v$ for each edge $v \to u$. As a result, Equation 1 is extended to the bidirectional case:

$$\mathbf{h}_u^{k+1} = \sigma \left( \exp_u( \sum_{v \in \mathcal{I}(u)} \tilde{\mathbf{A}}_{uv} \mathbf{W}^k \log_{\mathbf{x}'}(\mathbf{h}_v^k) + \sum_{v \in \mathcal{O}(u)} \tilde{\mathbf{A}}_{uv} \tilde{\mathbf{W}}^k \log_{\mathbf{x}'}(\mathbf{h}_v^k)) \right), \qquad (16)$$

where $\mathcal{O}(u)$ is the set of out-neighbors of $u$, i.e. $v \in \mathcal{O}(u)$ if and if only $u$ has an edge pointing to $v$. We use the mean candlestick price over a period of 8 hours in total as additional inputs to the network.

Table 4 shows the results. We compare against a baseline of inputting averaged 128-dimensional node2vec [20] features for the same time frame to an MLP classifier. We found that it helped if we only used the node2vec features for the top $k$ nodes ordered by degree, for which we report results here (and which seems to confirm our suspicion that the transaction graph is strongly hierarchical). In addition, we compare against using the autoregressive integrated moving average (ARIMA), which is a common baseline for time series predictions. As before, we find that Lorentz performs significantly better than Poincaré, which in turn outperforms the Euclidean manifold.

One of the benefits of using hyperbolic representations is that we can inspect the hierarchy that the network has learned. We use this property to sanity check our proposed architecture: if it is indeed the case that hyperbolic networks model the latent hierarchy, nodes for what would objectively be considered influential "whale" nodes would have to be closer to the origin. Table 5 shows the average norm of whale nodes compared to the average. For our list of whale nodes, we obtain the top 10000

addresses according to Etherscan[5], compared to the total average of over 2 million addresses. Top whale nodes include exchanges, initial coin offerings (ICO) and original developers of Ethereum. We observe a lower norm for whale addresses, reflecting their importance in the hierarchy and influence on price fluctuations, which the hyperbolic graph neural networks is able to pick up on.

## 5    Conclusion

We described a method for generalizing graph neural networks to Riemannian manifolds, making them agnostic to the underlying space. Within this framework, we harnessed the power of hyperbolic geometry for full graph classification. Hyperbolic representations are well-suited for capturing high-level structural information, even in low dimensions. We first showed that hyperbolic methods are much better at classifying graphs according to their structure by using synthetic data, where the task was to distinguish randomly generated graphs based on the underlying graph generation algorithm. We then applied our method to molecular property prediction on the ZINC dataset, and showed that hyperbolic methods again outperformed their Euclidean counterpart, as well as state-of-the-art models developed by the wider community. Lastly, we showed that a large-scale hierarchical graph, such as the transaction graph of a blockchain network, can successfully be modeled for extraneous prediction of price fluctuations. We showed that the proposed architecture successfully made use of its geometrical properties in order to capture the hierarchical nature of the data.

## Footnotes

[2]Other Riemannian manifolds such as spherical space are beyond the scope of this work but might be interesting to study in future work.

[3]https://deepchem.io/

[4]https://github.com/microsoft/gated-graph-neural-network-samples

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
