[Supplementary Material]

## A  Node Classification

As a sanity check of our implementation, we run Lorentz, Poincaré and Euclidean manifolds on three citation networks and compare the results with the original graph convolutional networks (GCN) implementation[1]. The results are shown in Figure 1. We find that the methods achieve comparable results, with the original GCN results reproduced. This shows that the implementation is correct. For node classification we should expect to see similar results for the different approaches.

|          | Cora           | Citeseer       | Pubmed         |
|----------|----------------|----------------|----------------|
| GCN      | 81.5           | 70.3           | 79             |
| Euclidean| $81.6 \pm 0.4$ | $70.9 \pm 0.4$ | $79 \pm 0.5$   |
| Poincaré | $80.8 \pm 0.4$ | $70.3 \pm 0.6$ | $78.2 \pm 0.6$ |
| Lorentz  | $81.9 \pm 0.3$ | $70.8 \pm 0.4$ | $79 \pm 0.8$   |

Table 1: Node Classification Accuracy on Citation Networks.

## B  Results of GCN and Euclidean Manifold

As another sanity check, we compare the results of the Euclidean graph neural networks when we remove the logarithmic map, the exponential map and normalization from the Euclidean manifold. Since we use the zero-vector as the fixed point to map into the tangent space, results should be similar. The results on ZINC are shown in Table 2. We observe that the Euclidean manifold and GCN achieve comparable results. We conclude that the logarithmic map, the exponential map and normalization do not have a significant effect on performance in Euclidean space.

|           | logP           |                |                |                |                |
|-----------|----------------|----------------|----------------|----------------|----------------|
|           | 3              | 5              | 10             | 20             | 256            |
| GCN       | $5.8 \pm 0.06$ | $4.9 \pm 0.04$ | $4.0 \pm 0.01$ | $4.0 \pm 0.03$ | $3.4 \pm 0.02$ |
| Euclidean | $6.7 \pm 0.07$ | $4.7 \pm 0.03$ | $4.7 \pm 0.02$ | $3.6 \pm 0.00$ | $3.3 \pm 0.00$ |
|           | QED            |                |                |                |                |
|           | 3              | 5              | 10             | 20             | 256            |
| GCN       | $22.8 \pm 0.10$| $16.5 \pm 0.06$| $12.9 \pm 0.08$| $10.3 \pm 0.02$| $6.8 \pm 0.06$ |
| Euclidean | $22.4 \pm 0.21$| $15.9 \pm 0.14$| $14.5 \pm 0.09$| $10.2 \pm 0.08$| $6.4 \pm 0.06$ |
|           | SAS            |                |                |                |                |
|           | 3              | 5              | 10             | 20             | 256            |
| GCN       | $19.5 \pm 0.03$| $16.4 \pm 0.06$| $16.1 \pm 0.01$| $9.6 \pm 0.08$ | $9.6 \pm 0.05$ |
| Euclidean | $20.5 \pm 0.04$| $16.8 \pm 0.07$| $14.5 \pm 0.11$| $9.6 \pm 0.05$ | $9.2 \pm 0.08$ |

Table 2:  Mean absolute error (low is good).

## C  Ethereum Dataset Construction Details

We use Geth[2] to download all blocks from August 7th, 2015 to September 9th, 2018. Using Web3.js[3], we parse the block data in order to extract the information of all non-zero transactions recorded on the blockchain. This creates a dataset of a little over 300 million transaction edges. For price information, we use the publicly accessible Poloniex API[4] to download four-hour candlestick chart data for the exchange rate of Ether/USDT (note that USDT is a token that is pegged to the USD price).

## D  Relative runtime

The computational overhead of using non-Euclidean manifolds is relatively minor, as can be seen in Table 1. The log and exp maps have simple closed-form solutions and require only computationally cheap operations (vector norms, elementwise division, cosh, etc.). The mapping between Lorentz and Poincaré is also efficient for the same reasons (see Eq. 12/13).

|  | Cora | Citeseer | Pubmed | ZINC |
|---|---|---|---|---|
| Lorentz | 1.014 | 1.048 | 1.03 | 1.285 |
| Poincaré | 1.041 | 1.125 | 1.083 | 1.34 |

Table 3: Relative runtime (defined as clock time of Poincaré or Lorentz divided by clock time of Euclidean).

## E  Graph Kernel Benchmarks

We additionally evaluate our methods on commonly-used graph kernel benchmark datasets [1]. On all datasets, we performed 10-fold cross validation and report the accuracy averaged over all folds in Table 4. We include these results in case they are useful for comparison in future work.

|  | D&D | Enzymes | Reddit | Collab | Proteins |
|---|---|---|---|---|---|
| Euclidean | $76.93 \pm 7.21$ | $43.83 \pm 10.3$ | $52.94 \pm 4.84$ | $83.01 \pm 4.53$ | $75.46 \pm 3.88$ |
| Poincaré | $75.89 \pm 8.53$ | $44.15 \pm 8.43$ | $51.86 \pm 6.86$ | $84.54 \pm 7.85$ | $73.64 \pm 4.64$ |
| Lorentz | $77.10 \pm 6.65$ | $44.83 \pm 8.14$ | $53.21 \pm 9.15$ | $88.96 \pm 3.37$ | $74.16 \pm 3.25$ |

Table 4: Graph Classification Accuracy on Graph Kernel Benchmarks.

## F  Empirical Results on QM8 and QM9

We further evaluated the performance of our methods on two widely used chemical prediction datasets, QM8 and QM9, shown in Table 5 and 6 respectively. We followed the same experimental settings as reported in Section 4.2. Lorentz performs substantially better than the alternatives on all properties.

| QM8 | DTNN | MPNN | GGNN | Euclidean | Poincaré | Lorentz |
|---|---|---|---|---|---|---|
| E1 - CC2 | $9.1 \pm 0.12$ | $8.4 \pm 0.13$ | $8.3 \pm 0.15$ | $8.2 \pm 0.18$ | $6.6 \pm 0.19$ | $6.2 \pm 0.1$ |
| E2 - CC2 | $9.5 \pm 0.12$ | $9.1 \pm 0.17$ | $8.5 \pm 0.14$ | $8.2 \pm 0.1$ | $11.8 \pm 0.16$ | $6.9 \pm 0.06$ |
| f1 - CC2 | $18.2 \pm 0.18$ | $16.7 \pm 0.3$ | $17.1 \pm 3.42$ | $17.9 \pm 0.17$ | $16.1 \pm 0.25$ | $15.6 \pm 0.25$ |
| f2 - CC2 | $36.7 \pm 0.48$ | $32.5 \pm 0.36$ | $34.2 \pm 3.42$ | $34.4 \pm 0.69$ | $31.8 \pm 0.44$ | $31.1 \pm 0.44$ |
| E1 - PBE0 | $9.2 \pm 0.15$ | $8.3 \pm 0.11$ | $8.7 \pm 0.1$ | $8.1 \pm 0.12$ | $6.3 \pm 0.17$ | $5.6 \pm 0.06$ |
| E2 - PBE0 | $8.6 \pm 0.15$ | $8.6 \pm 0.09$ | $8.8 \pm 0.15$ | $8.8 \pm 0.14$ | $6.8 \pm 0.21$ | $6.2 \pm 0.06$ |
| f1 - PBE0 | $15.1 \pm 0.27$ | $13.3 \pm 0.21$ | $13.1 \pm 0.2$ | $13.5 \pm 0.16$ | $12.1 \pm 0.2$ | $12.1 \pm 0.11$ |
| f2 - PBE0 | $28 \pm 0.45$ | $25.6 \pm 0.44$ | $25.8 \pm 0.36$ | $26.3 \pm 0.51$ | $24.8 \pm 0.34$ | $23.9 \pm 0.22$ |
| E1 - CAM | $8.6 \pm 0.16$ | $7.9 \pm 0.14$ | $7.5 \pm 0.11$ | $7.6 \pm 0.17$ | $7 \pm 0.1$ | $5.5 \pm 0.05$ |
| E2 - CAM | $8.4 \pm 0.15$ | $8.2 \pm 0.1$ | $8.8 \pm 0.16$ | $8.4 \pm 0.1$ | $6.6 \pm 0.19$ | $5.9 \pm 0.06$ |
| f1 - CAM | $16.9 \pm 0.2$ | $13.4 \pm 0.17$ | $14.3 \pm 0.17$ | $13.3 \pm 0.3$ | $11.8 \pm 0.27$ | $11.3 \pm 0.15$ |
| f2 - CAM | $32.9 \pm 0.39$ | $25.7 \pm 0.31$ | $25.8 \pm 0.41$ | $25.8 \pm 0.54$ | $25.3 \pm 0.3$ | $23.9 \pm 0.29$ |

Table 5: Mean absolute error. Scaled by 1000 for table formatting (low is good).

## Footnotes

[1]https://github.com/tkipf/gcn

[2]https://github.com/ethereum/go-ethereum

[3]https://github.com/ethereum/web3.js/

[4]https://poloniex.com/support/api/

## References

[1] Kristian Kersting, Nils M. Kriege, Christopher Morris, Petra Mutzel, and Marion Neumann. Benchmark data sets for graph kernels, 2016. `http://graphkernels.cs.tu-dortmund.de`.

| QM9 | DTNN | MPNN | GGNN | Euclidean | Poincaré | Lorentz |
|---|---|---|---|---|---|---|
| mu | $0.29 \pm 0.0072$ | $0.36 \pm 0.0053$ | $0.38 \pm 0.005$ | $0.37 \pm 0.0052$ | $0.30 \pm 0.0054$ | $0.24 \pm 0.0031$ |
| alpha | $0.64 \pm 0.00021$ | $0.61 \pm 0.00014$ | $0.43 \pm 0.0073$ | $0.42 \pm 0.0064$ | $0.41 \pm 0.0057$ | $0.31 \pm 0.005$ |
| HOMO | $4.9 \pm 0.081$ | $5.8 \pm 0.091$ | $8.9 \pm 0.02$ | $8.6 \pm 0.09$ | $7.4 \pm 0.09$ | $3.7 \pm 0.04$ |
| LUMO | $5.4 \pm 0.04$ | $6.9 \pm 0.03$ | $8.1 \pm 0.01$ | $7.9 \pm 0.013$ | $6 \pm 0.09$ | $3.5 \pm 0.04$ |
| gap | $6.8 \pm 0.063$ | $7.7 \pm 0.051$ | $8.4 \pm 0.01$ | $8.5 \pm 0.013$ | $7.6 \pm 0.09$ | $5.4 \pm 0.01$ |
| R2 | $17.2 \pm 0.22$ | $24.3 \pm 0.22$ | $14.9 \pm 0.18$ | $15.1 \pm 0.27$ | $15.6 \pm 0.28$ | $8.68 \pm 0.13$ |
| ZPVE | $1.9 \pm 0.018$ | $2.6 \pm 0.011$ | $0.6 \pm 0.01$ | $0.6 \pm 0.01$ | $0.7 \pm 0.01$ | $0.3 \pm 0.01$ |
| U0 | $2.43 \pm 0.013$ | $1.78 \pm 0.015$ | $0.25 \pm 0.0028$ | $0.25 \pm 0.0040$ | $0.21 \pm 0.0034$ | $0.16 \pm 0.0031$ |
| U | $2.01 \pm 0.018$ | $2.04 \pm 0.013$ | $0.42 \pm 0.0075$ | $0.40 \pm 0.0040$ | $0.35 \pm 0.0071$ | $0.19 \pm 0.0017$ |
| H | $1.64 \pm 0.012$ | $1.57 \pm 0.011$ | $0.34 \pm 0.0048$ | $0.35 \pm 0.0035$ | $0.30 \pm 0.0061$ | $0.14 \pm 0.002$ |
| G | $2.18 \pm 0.016$ | $1.24 \pm 0.012$ | $0.17 \pm 0.0023$ | $0.17 \pm 0.0034$ | $0.16 \pm 0.0017$ | $0.12 \pm 0.0022$ |
| Cv | $0.28 \pm 0.0032$ | $0.39 \pm 0.0032$ | $0.16 \pm 0.0028$ | $0.16 \pm 0.0030$ | $0.18 \pm 0.0025$ | $0.11 \pm 0.0020$ |

Table 6: Mean absolute error. Values for HOMO, LUMO, gap and ZPVE are scaled by 1000 for table formatting (low is good).