[Reviews · NeurIPS 2019]

Reviewer 1



Originality This paper is an interesting generalization of neural networks on Riemannian manifolds that are agnostic to space. It introduces the idea of working on the tangent space and using a pair of functions that are inverse to each other and move points from and to the tangent space. What is not clear in the paper is why we should use functions that move points to and from the tangent space and not any pair of inverse functions. If there is no theoretical reason and empirical would have been sufficient. It is also not clear why they claim generality on any network and they picked graph nets. Also the authors should have made more clear the difference with reference [15]. I think it is the fact that they extend to the hyperboloid vs the poincare disk, but it is not properly highlighted on the paper Clarity: The flow of the paper is very good and the method very clear. It would have helped if the message more crisp in the introduction. line 13: Reference [3] is very informative and backs your point really well. line 37: “Furthermore, we show that HGNNs are better able to model synthetic data” rephrase possibly, “are more suitable” line 38: Being able to model synthetic data is not really a contribution. Technically speaking I can always create synthetic datasets where the euclidean networks are more suitable line 50: “Using hyperbolic geometry for learning hierarchical representations of symbolic data was first proposed in [28], using the Poincaré ball model”, Try to rephrase this sentence to flow better line 79-81: If the method you develop works on any Riemmanian and it works on any space, then you should consider changing the title of the paper and make more general. line 84-96: This seems to be a bug of the proposed method. Let me explain what I mean here by the word bug. I argue that I don’t really need to use a log and exp function that corresponds to a Manifold of hyperbolic or a spherical metric. I can use any autoencoder where log is the encoder function and exp is the decoder function. Would I get better results? I think an experiment would have been worthwhile. line 138-141: I am not sure I agree with that. A regression model can take any features as input. I am not sure why this transformation is prefered. I have used it in data science projects successfully before, but I don’t see the mathematical connection here. Couldn’t you use mapping to a euclidean space and then do a simple regression? Why do you need centroid regression? Section 3.4: It is positive that authors mention these details that are critical to correct implementation of their algorithm Experiments: The methods that authors compared against are representative. I also liked the idea of synthetic data to prove the point. It would make also sense to use synthetic data generated by the first reference [1] and see if it can model it almost perfectly. I think it would have been better to pick different real datasets, like pubmed, cora that there is more rich comparison in the literature.

Reviewer 2



**SUMMARY * Goal: Extend the naturalness of hyperbolic spaces for encoding tree- and graph-structured data, observed with standard NNs, to Graph NNs. * Model innovation: Given: a graph and a Riemannian manifold M. Each node of the graph is annotated by a point of M. For all nodes in the graph, [carry out the standard linear operations for aggregating messages from neighboring nodes] in the tangent space T_x of the same point x on the manifold: the origin. This means the point of M annotating a node needs to be mapped to a vector in T_x, which is done by M's log map. The resulting vector in T_x is multiplied by a (relation-specific) learned weight matrix and a modified version of the graph's adjacency matrix. The new vector is then mapped back to a point of M by M's exp map. There it is subjected to a non-linear map from M to M. Classification and regression are performed on a transform of the set of final annotations of all graph nodes [137]. A set of n reference points in M is chosen [by a method not described -- update, the authors' reply indicates these are learned, which is nice, and should be made more explicit in the paper] and for each annotation a in M, a vector in R^n is produced by listing the distance of a to each reference point. Node-level prediction is performed by taking this transform of a node's annotation and computing: a linear functional of it (for regression); or the softmax of a linear transformation of it (for classification). For graph-level predictions, the same operations are applied to the average of the transforms of the annotations of all the nodes in the graph. * Complexity: No formal analysis is provided, although, non-technically speaking, the new method is quite complicated, especially in the Lorentz model where it is necessary to map over to the Poincaré ball to perform the nonlinearity, and then transform back. [See my comment in 5 below: I am fully satisfied by the authors' response.] * Tested models: This method is applied to two isometric hyperbolic manifolds, the Poincaré ball and the Lorentz model, which are compared to the Euclidean manifold of the same dimension. * Tasks: A synthetic task of determining which of 3 graph-generation algorithms produced the data [168]; properties of molecules (ZINC dataset) [190]; "predicting price fluctuations for the underlying asset of the Ethereum blockchain" [228] * Results: The Lorentz model is shown to outperform the Poincaré ball which outperforms the Euclidean case. **REVIEW * To a reader familiar with GNNs and Riemannian manifolds, the paper is clear. * The work seems quite original. * The experiments are presented clearly enough and the results seem rather good: e.g., on the task of predicting 3 molecular properties, with 256 dimensions, the SoTA/new model error values are: 3.2/2.4, 6.4/4.7, 9.1/7.7. ** TYPOs etc. * In line [113], presumably the norm intended is presumably the Euclidean norm; it would be worth making this explicit given that it's in a section devoted to defining other norms. The "unit ball" \mathcal{B} is not a unit ball in the space being constructed, of course. * In Eq 14 and following, presumably "d_{1j}" should be "\psi_{1j}"

Reviewer 3



The paper extends previous Hyperbolic network (embedding and attention) idea in GNN. It provides a direct formulation of Hyperbolic method in the Graph Neural Network setup. The derivation and utilization of Hyperbolic space in Riemannian manifolds are largely similar to previous Hyperbolic approaches in word embeddings and attention mechanisms. Experiments on syntactic data as well as several real-word datasets are provided. The current form of this paper is not ready for publishing because of several weaknesses. The first is that the techniques used to incorporate hyperbolic geometry are mostly the same with previous approaches---the main difference is that previous work applied the same set of techniques in embeddings and attentions, while this work put it in GNN. Technical wise the contribution is not significant. The second is that the proposed method does not show convincing results on the syntactic dataset constructed for hyperbolic GNN methods. There is not much insight to learn from the syntactic experiments. The effectiveness of the proposed method on real-world datasets are also not significantly better than GCN, which is already a weak baseline, consider the rapid process in GNN research.

Reviewer 4



The authors present an adaptation of Graph Neural Networks which learns node representations on Riemannian manifolds-- and more precisely, hyperbolic spaces, which have negative curvature. The underlying motivation for this approach arises from the recent successes of Geometric deep learning and the use of hyperbolic spaces for embedding learning in a variety of applications. In this paper, the authors describe the manifold that they embed their representations into (Poincarre and Lorentz), detail the corresponding updates, and show the performance of their algorithm for general graph classification tasks on one synthetic experiment and two concrete applications. These experiments seem to highlight the benefits of learning non-Euclidean representations of the data, whose geometry seems more suitable to capture fundamental properties of the data, such as clustering or hierarchies, especially in low-dimensional settings. The paper is clearly written and easy to follow. The experiments are rather well described--although some key details are very blurry, and seeing the actual code might have lifted some shadow areas--, and the “sanity-check” experiments provided in the supplementary material provide a sound basis for the authors' discussion. The discussion around this submission can be organized along the following axes: (1) From the novelty perspective, this paper is extremely reminiscent of the work by Gulcehre et al (ICLR 2019)[1]--although it is not cited here. Indeed, in their ICLR 2019 paper, Gulcehre et al propose to learn hyperbolic embeddings for Graph Neural Attentions mechanisms: their goal is to learn relationships in graphs. Attention weights are defined as distances in hyperbolic spaces, and the message passing updates are defined by taking the Einstein midpoint. This message-passing process yields representations of the nodes that can then be used for any downstream ML task. The main difference with the current paper lies in this message-passing step: in this current submission, learnable operations are performed in the Manifold's tangent space—which is Euclidean, so brings us back to a familiar setting where we can perform linear and learnable operations-- rather than the manifold itself. As such, the messages are less “guided”, since they do not use the Einstein midpoint, but rely on using successively using the manifold’s exponential and logarithmic maps to go to and fro the Euclidean and hyperbolic spaces. This is a new and interesting viewpoint. It would be interesting to dig deeper into what this entails from a theoretical perspective, especially since the tangent space here is always taken with respect to the origin… How does this affect the results? The paper might thus benefit from highlighting further this difference with this competing hyperbolic method, but it seems that the mechanism itself still remains sufficiently novel. (2) Another interesting contribution is the authors's use of these embeddings for node-wise and graph-wise classification tasks. To adapt regression and classification methods to handle their hyperbolic embeddings, the authors introduce a set of centroids that act as landmarks and summarize the structure of the graph's embeddings: each node is characterized in Euclidean space by the distance of its embedding to the centroids (where I assume the distance is the hyperbolic one-- in which case, what does the term "centroid" mean? Is landmark a more correct characterization? Or are these actual centroids, in which case, which definition of "mean" or center is used? The paper would also perhaps benefit from detailing how these centroids are selected: are they fixed? Are they random? Do they change for each graph?). The assumption though is that the hyperbolic distances with respect to those fixed manifold landmarks characterize better the density of the data than distances between vanilla Euclidean embeddings, and as such, will be serve as better data input… However, this places the method somehow on-par with other "similarity-learning" methods. As such, for a more thorough understanding of the authors' method, this classification task should perhaps be compared with other such methods: in the case of node-identified graphs for instance, one could use the WL kernel to compute similarities between nodes, and taking a set of fixed nodes as "centroids", perform classification similarly as in this paper... (3) From the experiments perspective: the experiments might be a little weak to show clearly the benefits of using such hyperbolic methods in light of other methods: a. For the synthetic experiments: the experiments seem to indicate the superiority of HGNN compared to Vanilla GNNs. However, the task itself might be very simple, and easily achievable with non-neural network based methods. For instance, in the 3 dimensional setting, are simple statistics (like the mean, standard deviation, and max degree over the graph--all of these perhaps normalized by the number of nodes) sufficient to represent the graph and achieve good classification performance? If this simple method performs as well as HGNN or any message-passing based method, then why learn such a complicated model when logistic regression on vanilla features could suffice? b. Similarly, for the Molecule dataset, the authors choose to compare their method with other “state-of-the-art” deep learning competitors. However, Weisfeiler-Leman kernels have proven extremely accurate on molecule prediction tasks, so yet again, it remains to show that neural network based models (hyperbolic or not) are significantly better than other methods… (see [2] for instance). Moreover, in table 3, it seems that the authors have used the high dimensional version of the hyperbolic space to compare with other SOTA approaches… What were the dimensions used in competing methods? To sum up, the message-passing update mechanism proposed by the authors is a nice and interesting contribution which seems sufficiently novel to distinguish it from other hyperbolic graph neural network methods already proposed in the literature. Perhaps the main drawback of the paper is a lack of definite direction, neither emphasizing the theoretical novelty of the method nor providing extensive experiments to justify it: while the results presented by the authors would make it interesting to dig deeper in the theory behind these updates, the experiments might still be a little lacking, as they focus on only showing the hyperbolic embeddings success with respect to other neural-network-based methods, not withstanding the rich literature on non-DL based methods for the same task. It would also be beneficial to see the performance of their method on a wider number of tasks and dataset. [1] __HYPERBOLIC ATTENTION NETWORKS__, Caglar Gulcehre, Misha Denil, Mateusz Malinowski, Ali Razavi, Razvan Pascanu, Karl Moritz Hermann, Peter Battaglia, Victor Bapst, David Raposo, Adam Santoro, Nando de Freitas [2]__ Weisfeiler and Leman Go Neural: Higher-order Graph Neural Networks__, Christopher Morris, Martin Ritzert, Matthias Fey, William L. Hamilton, Jan Eric Lenssen, Gaurav Rattan, Martin Grohe

[Author Response · NeurIPS 2019]

**Reviewer 1**  We thank the reviewer for the insightful and encouraging comments.

*Regarding the choice of logm/expm vs. any other pair of invertible functions* Logarithmic and exponential maps are the principled way to define a mapping to a locally Euclidean space on Riemannian manifolds (and are, for instance, used in the definition of geodesics on the manifold). For our purpose, they have the additional benefit that they are parameter-free, whereas an autoencoder would have to be learned. On the manifolds that we are considering, they also have closed-form solutions, are differentiable, and are computationally efficient. Please see also Table 2, which lists additional experiments with a 1- and 2-layer autoencoder and which illustrates the advantages of our method.

*Regarding the difference to hyperbolic NNs* The work of Ganea et al. is indeed related to our work and we tried to highlight these connections. Important differences to our work include: (1) Our formalism is more general as we can handle any Riemannian manifold with differentiable logm/expm. Our formalism allows us also to easily extend GNNs to the Lorentz model which provides important benefits as shown in our experiments. (2) We propose a generalization of message-passing GNNs to Riemannian manifolds whereas the work of Ganea et al. is focused on classic neural-networks such as RNNs. (3) We propose a novel centroid-based method to classify embeddings in Riemannian NNs.

*Regarding the benefits of our centroid-based classification* There exist no general isometric mappings between Hyperbolic and Euclidean space. Moreover, a parametric function to learn a task-specific mapping between both spaces would (in our experience) likely require many parameters and training examples to learn the complex relations. For this reason, we chose a "non-parametric" centroid-based approach which is computationally efficient and flexible.

|         | Cora  | Citeseer | Pubmed | ZINC  |
|---------|-------|----------|--------|-------|
| Lorentz | 1.014 | 1.048    | 1.03   | 1.285 |
| Poincaré| 1.041 | 1.125    | 1.083  | 1.34  |

Table 1: Relative runtime (defined as clock time of Poincaré or Lorentz / clock time of Euclidean)

|            | logP          | QED           | SAS           |
|------------|---------------|---------------|---------------|
| 1-layer AE | $3.2 \pm 0.03$ | $6.7 \pm 0.01$ | $9.5 \pm 0.05$ |
| 2-layer AE | $3.3 \pm 0.04$ | $6.9 \pm 0.02$ | $9.6 \pm 0.08$ |
| Log and exp| $2.4 \pm 0.02$ | $4.7 \pm 0.0$  | $7.7 \pm 0.06$ |

Table 2: Lorentzian GNN (dimension 256) on ZINC using either an autoencoder or log & exp maps.

**Reviewer 2**  We thank the reviewer for the insightful and encouraging comments.

*Regarding code release* Thank you for raising this issue. We do indeed plan to fully open-source our code on Github.

*Regaring the time complexity of our method* The computational demand of our method is similar to standard GCNs. Logm and Expm have simple closed-form solutions and require only computationally cheap operations (vector norms, elementwise division, cosh, etc.). The mapping between Lorentz and Poincare is also efficient for the same reasons (see Eq. 12/13). Table 1 shows the relative runtime performance for Lorentz and Poincare GNNs compared to Euclidean GNNs. It can be seen that the computational overhead is low and ranges between 1.01-1.34 times the Euclidean model.

*Regarding reference points in centroid classification* The reference points are actually learned jointly with the GNN using backprop. This enables the model to infer relevant areas in the embedding jointly with the GNN.

**Reviewer 3**  We thank the reviewer for the comments which will help us improve our paper. However, we respectfully disagree with several aspects of the review:

*Regarding experiments on synthetic datasets* Please note that a Lorentzian GNN with only 10 dimensions outperforms the Euclidean GNN with 256 dimensions. The improvements are statistical significant as can be seen from the reported error bars. Furthermore, there is a large improvement over Euclidean GNNs in low-dimensions (F1 of 94.1 vs 77.2 in d3) which further illustrates the benefits of Hyperbolic geometry for this task. The goal of these synthetic experiments is to evaluate the ability of models to capture graph topology and our results show clearly that Hyperbolic GNNs outperform Euclidean GNNs on this task in absolute performance and additionally have large benefits in smaller dimensions.

*Regarding experiments on real-world datasets* The task of molecular prediction is a common benchmark for GNNs. For this task, we do not only compare to Euclidean GNNs, but also to current state-of-the-art methods for molecular prediction, i.e., MPNN, DTNN, GGNN. Our evaluation in Table 3 of the paper shows clearly that we get statistical significant improvements using our approach. In our experiments on Blockchain data, we furthermore compare to ARIMA which is a commonly used and strong baseline for this task. It is therefore unclear to us why the review criticizes a lack of state-of-the-art baselines and/or significant improvements as we clearly show both.

*Regarding technical contributions* Please see the response to R1 for a discussion of novel contributions compared to hyperbolic NNs. Furthermore, our paper is the first to study the benefits of hyperbolic geometry in the context of GNNs. Our experiments show the advantages of this approach both with regard to representational efficiency (strong performance of low dimensional embeddings) and generalization ability (state-of-the-art results on synthetic and real-world data). We therefore believe that our manuscript makes valuable contributions to advance the state-of-the-art for GNNs.

[Meta-Review · NeurIPS 2019]

After reading the paper, reviews, and rebuttal, and after discussing with the reviewers, I solicited a fourth reviewer to weigh in. The consensus is that the core contribution is an adaptation of hyperbolic neural networks to handle structured (graph) inputs, with some subtleties in terms of how the constraints of hyperbolic space are maintained. The paper is quite similar to recent ICLR '19 work on hyperbolic attention networks (see R4 comments); this is the related work R3 was referring to, but unfortunately did not explicitly reference in his original review. Still, it is quite surprising that this paper is not referenced in the original submission since it has been on arXiv and OpenReview for many months and would have turned up in a cursory Google Scholar search. The authors must make clear the connections to this work in the final proceedings. R4 does identify some points of departure from the ICLR paper, and suggests that there is enough novelty in this submission to warrant publication in NeurIPS.